# An Innovative Concept of a 3D-Coded Aperture Imaging System Proposed for Early Breast Cancer Detection

**DOI:** 10.3390/diagnostics12102529

**Published:** 2022-10-18

**Authors:** Khalid Hussain, Mohammed A. Alnafea, M Iqbal Saripan, Djelloul Mahboub, Rozi Mahmud, Wan Azizun Wan Adnan, Dong Xianling

**Affiliations:** 1Department of Computer and Communication System Engineering, Universiti Putra Malaysia, Serdang 43400, Selangor, Malaysia; 2Department of Radiological Sciences, College of Applied Medical Sciences, King Saud University, Riyadh 11433, Saudi Arabia; 3Physics Department, University of Hail, Ha’il 81451, Saudi Arabia; 4Faculty of Medicine and Sciences, Universiti Putra Malaysia, Serdang 43400, Selangor, Malaysia; 5Department of Biomedical Engineering, Chengde Medical University, Chengde City 067050, China

**Keywords:** coded-aperture imaging, anthropomorphic breast phantom, mosaic MURA mask, gamma camera, mask/antimask, 3D image reconstruction

## Abstract

Coded Aperture (CA) imaging has recently been used in nuclear medicine, but still, there is no commercial SPECT imaging camera based on CA for cancer detection. The literature is rich in examples of using the CA for planar and thin 3D imaging. However, thick 3D reconstruction is still challenging for small lesion detection. This paper presents the results of mosaic modified uniformly redundant array (MURA) mask/antimask CA combined with a maximum-likelihood expectation-maximization (MLEM) algorithm. The MLEM is an iterative algorithm applied to a mosaic MURA CA mask/antimask for 3D anthropomorphic breast phantom reconstruction, slice by slice. The difference between the mask and the antimask suppresses the background noise to enhance the quality of reconstructed images. Furthermore, all reconstructed slices are stacked to form a 3D breast phantom image from single-projection data. The results of phantom reconstruction with 8 mm, 6 mm, 4 mm, and 3 mm lesions are presented. Moreover, the proposed SPECT imaging camera can reconstruct a 3D breast phantom from single-projection data of the patient’s scanning. To assess the quality of lesions in the reconstructed images, the contrast-to-background ratio (CBR), the peak signal-to-noise ratio (PSNR) and mean square error (MSE) were measured.

## 1. Introduction

Nuclear medicine imaging methods are actively used for the diagnosis of cancer at the early stages. The single-photon emission tomography (SPECT) and positron emission tomography (PET) are the state-of-the-art nuclear imaging modalities used for cancer diagnosis. However, SPECT imaging is cheaper and widely used in modern hospitals. Among the several types of cancers, breast cancer is the most prominent cancer in women. According to GLOBOCAN 2020, female breast cancer is the most commonly diagnosed cancer, with 11.7% of new cases and a death rate of 6.9% [1]. To reduce the mortality rate of breast cancer, a lot of research has been done[1,2,3,4,5,6]. The SPECT is a non-invasive molecular imaging modality that employs parallel holes for the collimation of radioactive emissions from the patient’s body towards a gamma camera.

Coded Aperture (CA) imaging techniques were developed as a substitute for parallel hole collimators to improve spatial resolution and increase sensitivity while keeping the patient dose at a minimum, but still, there is no practical SPECT system in use based on coded aperture. The coded aperture is a generalization of pinhole imaging which can increase the signal-to-noise ratio (SNR) while maintaining the angular resolution of the single-pinhole camera [7,8]. A pinhole camera is an established direct imaging technology that produces high-resolution, high-quality images but has a limited field-of-view (FoV). In the context of nuclear medicine imaging, limited angle acceptance and poor geometric performance result in long acquisition times as the imaging process is inefficient due to the small size of a single pinhole. The SNR can be increased by a big opening of the pinhole while a big opening will decrease the angular resolution. Therefore, there is always a trade-off between resolution and SNR or sensitivity (the number of photon counts passing through the pinhole). The fundamental idea is to overcome photon scarcity by opening a large number of small pinholes on a thin shield rather than on a thick one, making it semi-transparent. The high spatial resolution is attained using a large value of magnification factor, whilst high sensitivity is achieved by drilling a large number of pinholes in the coded mask. Moreover, CA can increase the signal-to-noise ratio. The multiplexed image decoding technique uses the correlation properties of URA [9]. CA imaging methods can be used in three-dimensional (3D) imaging in addition to two-dimensional (2D) planar imaging. In 3D imaging, the projected image of CA contains depth information on the distribution of the radioactive elements [10], which can be obtained by using multiple cameras or rotating the camera around the patient. However, thick objects create artifacts because only one plane of the object can be focused at a time by a one-view coded-aperture camera, and all other planes are out of focus and add artifacts to the image. The near-field artifacts arise due to the intensity modulation of particles and are discussed by[11,12,13,14]. The shape of the mask also affects the quality of the reconstructed image. Among the several design patterns of coded masks, Uniformly Redundant Arrays (URA) or MURA are reported to be the best choice with minimum side lobes. Moreover, the mosaic MURA has an encouraging spatial resolution, contrast, and SNR. The comparison was made among MURA mask/antimask, mosaic MURA mask and mosaic MURA mask/antimask techniques. Among three techniques, the mosaic MURA mask antimask showed better performance; therefore, it is used for the simulation of 3D anthropomorphic breast phantom. This mosaic MURA mask/antimask technique reduces the noise of a modified uniformly redundant array mosaic. There are several modern Monte Carlo codes used for medical imaging, including MCNP [15], PENELOPE [16], Geant4 [17], and its derivative GATE [18].

This paper simulated the realistic anthropomorphic female small-sized breast phantom for different tumor sizes using GATE Monte Carlo simulation software. The tumor was placed at half of the depth of the breast, and the diameter of the tumor varied from 8 mm to 3 mm. The M-MLEM, an iterative reconstruction algorithm, was used for image reconstruction [19]. For the 3D reconstruction of the breast phantom, the image slices are reconstructed with a mask and 90 degrees rotated mask, which is also the antimask. The difference between reconstructed slices of the mask and the antimask minimized background noise and gave promising results. In addition, we obtained a 3D reconstructed image from a single scanning of the object instead of reconstructing it from multiple projections by rotating the camera around the patient. The contrast-to-background ratio, peak signal-to-noise ratio, and mean square error of the reconstructed images are measured and reported in Section 3, the results section.

## 2. Materials and Methods

### 2.1. Geometrical Structure

The simulated geometrical structure consists of a 3D anthropomorphic female breast phantom, the mosaic MURA coded-aperture mask, a scintillation detector, and a backscatter compartment. However, the mask and detector are both shielded for uncollimated radiations using tungsten material of 1.5 mm thickness. The half-ellipsoidal anthropomorphic breast phantom used in this research is of small size with dimensions given in Figure 1. The detector is made of sodium iodide (NaI) with an area of 66.4 × 66.4 cm^2^ and is filled with NaI material with a density of 3.67 g/cm^3^ and a thickness of 0.9525 cm. The backscattering compartment was made of Pyrex glass and had dimensions of 66.4 × 66.4 × 6.8 cm^3^. The mosaic MURA mask has dimensions of 16.6 × 16.6 × 0.15 cm^3^ with a fixed hole size of 1 mm^2^ and is made of tungsten, which has almost 99.4% attenuation to normal incident gamma rays of 140 keV. The geometric resolution and the field-of-view at the center of the breast depth for breast phantom simulation is 1.33 mm and 22.133 cm, respectively. The FoV is the ratio of the detector size and the object magnification, which vary by varying the magnification factor. The FoV will decrease by increasing the magnification factor and vice versa. The magnification factor at the center of the anthropomorphic breast phantom is 4, and it varies from 3.5 to 4.75 with respect to the depth of the slice. The slice depth is 1 mm, and the distance from the mask varies from 120 mm to 80 mm from the farthest slice to the closest slice. The distance from the mask plane to the detector plane is fixed and is 300 mm.

To simulate a clinical examination for breast cancer in the prone position, a half-ellipsoidal anthropomorphic breast phantom is used as an approximation of the realistic breast [20,21]. The phantom consists of ductal networks, adipose tissues, glandular tissues, skin, and the lesion placed in the center of the breast phantom, as shown in Figure 1. The atomic compositions of all breast tissues were taken from ICRUP report 44 [22]. The simulated geometrical structure developed using GATE is given in Figure 2. The proper shielding with high-density material near 100% attenuation towards simulated particles at a given energy is very important, otherwise, the image reconstruction may suffer degradation due to false counts. The tungsten material of 1.5 mm thickness is used for shielding the mask as well as the detector to stop the uncollimated radiations. The patient is kept in a chamber where the only breast part is visible out of the chamber for radiation emission. In this way, the whole body of the patient is shielded except the breast, but there are still radiations coming out from the chest just behind the breast. Although, the background radiations’ contribution is very important, which should be addressed in the imaging process rather than the reconstruction process. In this paper, we are only providing the proof of concept that the 3D reconstruction of the breast phantom is possible with coded-aperture imaging. Therefore, only the breast is modelled for the simulation and adding the contribution of the chest and background will be the subject of future research. Right now, we are interested in reconstructing a 3D anthropomorphic breast phantom using a coded-aperture mask. Therefore, the radiations’ contribution other than the breast is eliminated and only radiations coming out from the breast are considered.

### 2.2. Anthropomorphic Breast Phantom Activity

The activity of the breast should be allocated appropriately in order to make the simulation as near to the actual reality as possible. Patients are often administered with a total dose of 20 *m*Ci of ^99m^Tc for breast imaging, and then patients are asked to relax for 10 min to enable the Tc 99 m to disperse thoroughly. Although the activity is distributed differently in the different organs of the body, a rough volume ratio calculation should give us a good estimation of the activity inside the breast. The volume ratio of the simulated breast to an average body volume is 134 mL/80 L. This led to a breast activity of 33.5 µCi, hence a total number of emitted gamma rays of circa 750 million during the acquisition time of 10 min for the breast phantom with 4 pi solid angle emission, similar to the real angle of the patient dose distribution during the physical examination. The breast phantom is voxelized with a voxel volume of 0.125 mm^3^, whilst the voxel size along each direction is 0.5 mm. Each breast phantom voxel simulates an activity of 1.15 Bq for 10 min to yield 750 million events from the breast and tumor voxels of the anthropomorphic breast phantom. The tumor activity is taken to be ten times more than the activity of the breast, keeping in mind the tumor-to-background ratio of 10:1. Moreover, the activity in all breast voxels is uniformly distributed as per the TBR.

### 2.3. Energy Windowing and Blurring Process

The sensitive detector is designated as a part of scanner geometry, and the particle interactions are recorded in this region. The hits and singles are generated from these interactions that occur inside the sensitive detector. The singles give us information about the particle’s start location, energy deposited in the detector, and the positional coordinates of the particles at the time of energy deposition. The GATE digitizer [23] has built-in modules that are used for setting the energy window and blurring the energy and position. In the simulations, energy resolution of 10% and spatial resolution of 1 mm, a smaller value than geometric resolution is used for energy and position blurring using Gaussian blurring. The energy window of 10% is used for the maximum 140 keV of ^99m^Tc in the range of 126 keV to 156 keV.

### 2.4. Near-Field MURA Imaging Design Issues

The design of a mask pattern is a challenging process to select the best suitable pattern with the right size [24]. The coded-aperture imaging mathematics demonstrates that apertures with excellent correlation properties produce ideal point-spread functions, but this is only true in far-field examples. In near-field imaging design, where rays within the same point source in the object cannot be regarded as parallel, several variables can cause divergence from the ideal. The finite thickness of the object and the modulation of the intensity of the aperture projection on the detector due to variation in the angle of incidence of gamma particles are the most serious factors which are challenging issues while designing the geometry of the coded mask. Another design question is to achieve the maximum throughput of the mask to keep the patient dose as minimal as possible. The MURA pattern has a 50% open area transparent to radiation and is hence considered to be the optimum solution for clinical application, which minimizes the patient dose. Moreover, the variation in the angle of incident photons adds intensity modulation to the projected images on the detector, which causes degradation in the quality of reconstructed images [13]. Therefore, an improved decoding algorithm is also required to overcome these artifacts along with geometrical design. Our technique of subtraction combined with the MLEM decoding algorithm written in house has a remarkable performance in achieving delta correlation function on MURA-CA with minimum background noise.

### 2.5. Coded Mask Design

In the perspective of image performance, the auto-correlation function of the mask pattern should be a delta function. Different designs of the mask, i.e., binary Fresnel Zone Plate, sinusoidal zone plate, L- and X-shaped array, non-redundant array, uniformly redundant array, and MURA, were investigated by Alnafea [25]. It has been demonstrated that the Fresnel Zone Plate (FZP), non-redundant, X- and L-shaped designs have side lobes which cause image artifacts when used for reconstruction; thus, these are not considered for clinical application as an optimal design choice. However, the cyclic difference sets such as URAs and MURAs have been explored as the most promising patterns of coded aperture with high transmission characteristics (have 50% close/open area, i.e., the area transparent to gamma particles and opaque to gamma particles is the same) and flat (have zero) side lobes. In this research, as shown in Figure 3, we used a mosaic MURA mask built from a basic 83 × 83 binary pattern of the modified uniformly redundant array, which was originally developed and expanded by Fenimore and Cannon [26]. Figure 4 represents the response function of the mosaic mask.

The binary mask array is a pattern of 0s and 1s, where zeros correspond to an opaque area, while ones represent the holes that are transparent to gamma particles to record flux on a detector. The box-shaped hole facet has dimensions of 1 mm × 1 mm each and is mounted into a fixed holder of 66.4 cm × 66.4 cm.

The mosaic mask is initially developed from an 83 × 83 binary MURA pattern, as described by [27]. The mosaic MURA mask/antimask technique results better than MURA mask/antimask and mosaic MURA mask reconstruction. The comparison of MURA mask/antimask, mosaic MURA mask and mosaic MURA mask/antimask for breast phantom reconstruction with an 8 mm lesion has been given in Table 1, in Section 3, Results and Discussion. Due to the better performance, the mosaic MURA mask/antimask is used in this study for breast phantom reconstruction with lesions of sizes ranging from 8 mm, 6 mm, 4 mm to 3 mm.

The response function of mosaic MURA has one full delta function corresponding to the full copy of the original reconstructed object and three partial deltas that carry partial copies of the original simulated objects, and it is advantageous in enhancing the contrast of the reconstructed image.

### 2.6. Near-Field Artifacts in Coded Aperture Imaging

The CA imaging was first established for far-field imaging, in which all incident rays were parallel. The distance from the object to the mask is limited in near-field CA imaging, as used in nuclear medicine. There are significant differences between near-field and far-field imaging, as in far-field imaging, the rays are parallel to the detector, while in near-field coded-aperture imaging, the rays make an angle that causes intensity modulation, which degrades the reconstructed image. Therefore, we corrected the projection before reconstruction and obtained a better contrast of the image. Our implemented near-field correction MATLAB algorithm is adoptive for any shape of source and removes the near-field artifact significantly [12,13,25]. The size of the mask shadow is enlarged by the magnification factor. Moreover, the object size, when reconstructed, is zoomed by the ratio between the distance from the mask to the detector with the distance from the object to the mask.

### 2.7. Point-Source Simulation

First, we simulated the mosaic coded-aperture mask using a point source with 10^8^ particles. The radioactive isotope of ^99m^Tc that has an energy of 140 keV is used for the point-source simulation, and an energy window of 10% is considered. We corrected the projection for near-field correction, and it can be seen from Figure 5 that the raw projection of the point source has improved after near-field correction (NFC); therefore, the contrast-to-background ratio of the reconstructed point-source object using the MLEM reconstruction method is improved by 13.65% from 82.19 to 94.24 after applying our near-field correction method. After testing the mosaic mask on the point source, we used this mask for anthropomorphic breast phantom simulation and achieved encouraging results that are reported in the Results and Discussion.

### 2.8. Breast Phantom with Tumor

The half-ellipsoidal anthropomorphic breast phantom has been simulated for various sizes of tumor diameter under the fixed tumor depth and the tumor-to-background ratio (TBR). The simulated sizes of tumors are 8 mm, 6 mm, 4 mm, and 3 mm in diameter, and the fixed TBR value is 10:1. The tumor part of the breast phantom has an activity of 10 times higher than the activity of the rest of the breast. The breast-imaged area has approximately uniform activity in all the breast and tumor tissue as per the tumor-to-background ratio.

### 2.9. Image Reconstruction Methods

A 3D anthropomorphic breast phantom image can be represented as a stack of 2D slices in the z-direction. The depth of each slice will give a different magnification factor which is used for the image reconstruction using MLEM iterative reconstruction algorithm. The maximum-likelihood expectation-maximization in 3D SPECT reconstruction was used by Hong et al. [12]. For 2D planer image, Equation (1), and for 3D image reconstruction, Equations (2) and (3) are used for 200 iterations.
(1)$$f^{\left(k+1\right)}\left(x,y\right)=f^{\left(k\right)}\left(x,y\right)\left[\frac{p\left(x,y\right)}{f^{\left(k\right)}\left(x,y\right)\times h\left(x,y\right)}⨂h\left(x,y\right)\right]$$
(2)$$f^{\left(k+1\right)}\left(x,y,z\right)=f^{\left(k\right)}\left(x,y,z\right)\left[\frac{p\left(x,y\right)}{f^{\left(k\right)}\left(x,y,z\right)\times h\left(x,y,z\right)}⨂h\left(x,y,z\right)\right]$$
(3)$$f_{rot90}^{\left(k+1\right)}\left(x,y,z\right)=f_{rot90}^{k}\left(x,y,z\right)\left[\frac{p\left(x,y\right)}{f_{rot90}^{k}\left(x,y,z\right)\times h\left(x,y,z\right)}⨂R_{rot90}\left(\;h\left(x,y,z\right)\right)\right]$$

For 3D reconstruction, the modified maximum-likelihood expectation-maximization, an iterative reconstruction algorithm with a mosaic mask, and the 90 degrees rotated mask, was used. The difference between iteratively reconstructed slices with the mask and the antimask combined with MLEM reconstruction gives a promising image by minimizing the background noise. In Equations (1)–(3), the f is the estimate of the original image, which continuously improves with the number of iterations, while the *p*(*x,y*) is the raw projection of the breast phantom on the detector, and the *h*(*x,y*) and R_rot90_ (*h*(*x,y*)) represent the mask, and 90 degrees rotated mask. The z is the depth of the slice, and it varies from −20 to +19 with a 1 mm slice depth for a total number of 40 slices. The reconstructed breast slices using a mask, antimask, and the subtraction of the mask and antimask with tumor at a depth of 12.5%, 25%, and 50% are shown in Figure 6.

To emphasize the performance of the mosaic MURA mask/antimask technique used in this paper, a comparison is made among MURA mask/antimask, mosaic MURA mask, and mosaic MURA mask/antimask by simulating the breast phantom with an 8 mm tumor size. From the reconstructed images of the three techniques, the mosaic MURA mask/antimask performs better than the other methods, as shown in Figure 7.

The reconstructed image using the MURA mask/antimask has lower contrast than the mosaic MURA. Secondly, the mosaic MURA mask reconstructed image has better contrast, but it also has more noise in the background, eventually degrading the image quality. Finally, the mosaic MURA mask/antimask has enhanced contrast with reduced noise. Moreover, the difference between the mosaic MURA mask/antimask reconstructed image has minimized the background noise by subtracting the antimask image from the mask image and thus improved the image quality. Therefore, the mosaic MURA mask/antimask is used for the simulation of breast phantoms with different lesions in this research.

Due to the mosaic nature of the MURA mask, the reconstructed images also have partial copies along the axial directions and at the corners other than the full reconstructed image in the center, as can be seen from Figure 5. These partial copies can be added properly with the central full reconstructed image to enhance the contrast, which will improve the image quality. For the comparison, the difference of contrast for the 8 mm lesion, for the central reconstructed image, and for the central part added with partial copies were calculated as 0.6032 and 0.6750, respectively. Moreover, when the central image is combined with partial copies of the axial directions, the contrast is improved by 11.23%, with the expense of a 0.52% increase in MSE and a 0.029% decrease in PSNR. The planer view of the 3D reconstructed breast phantom with an 8 mm lesion is shown in Figure 8 below.

## 3. Results and Discussion

The experiments in this section are carried out using a mosaic MURA mask for the simulation of an anthropomorphic, realistic female breast phantom of small size with different tumor diameters. For image reconstruction, a maximum-likelihood expectation-maximization with a mask and the antimask was used.

To evaluate the performance of the mosaic MURA mask/antimask technique, a comparison was made among the reconstructed images of the MURA mask/antimask, mosaic MURA mask, and mosaic MURA mask/antimask. The results are presented in Table 1.

The mosaic mask CA imaging system, together with MLEM reconstruction, gives a 3D reconstructed image from a single projection that can be obtained by scanning the patient from a single angle, unlike existing SPECT scanning systems where the camera is rotated around the patient through a 360° angle of rotation. In the first simulation, the 8 mm diameter lesion was used, and the quality of the reconstructed image was assessed. After obtaining promising results from the 8 mm diameter lesion, subsequently, the 6 mm, 4 mm, and 3 mm diameter lesions were simulated. In all simulations, the tumor depth, the tumor-to-background ratio (TBR), and activity were fixed. The breast phantom and tumor were simulated for TBR of 10:1 and the activity of 33.5 µCi for 10 min acquisition time, totaling the 750 million particles with 360 degrees of emission angle, as it is in the actual physical examination.

The quality of the reconstructed breast phantom with different sizes of tumor is determined by measuring the contrast-to-background ratio (CBR), peak signal-to-noise ratio (PSNR) and mean square error (MSE) for all the simulated images. The reconstructed 3D breast phantom images with a tumor of 8 mm, 6 mm, 4 mm, and 3 mm are shown in Figure 9. The reference images in measuring the PSNR and MSE were source images of actual breast phantom with the given size breast tumor.

Figure 10 depicts the tumor properties in the reconstructed images of the anthropomorphic breast phantom. We calculated the contrast-to-background ratio, peak signal-to-noise ratio (PSNR), and mean square error (MSE) of all the reconstructed images using the method described by [28].

The results of the 3D reconstructed breast phantom are presented. The values of the peak signal-to-noise ratio and the contrast-to-background ratio are higher for the bigger tumor and gradually decrease as the diameter of the tumor decreases. Moreover, the mean square error for an 8 mm tumor is minimum, and it increases as the tumor size reduces.

## 4. Conclusions

The performance of the mosaic MURA mask for a 3D anthropomorphic breast phantom with a tumor of variable sizes was evaluated using the MLEM iterative reconstruction algorithm. Firstly, we reconstructed 3D images from single-projection data using mosaic MURA mask CA, using MLEM reconstruction. Secondly, the proposed MLEM method employs dual reconstruction from the mask and the antimask and has the ability to significantly minimize the background noise by subtracting the mask reconstructed image from the antimask reconstructed image. The results indicate that the mosaic MURA mask combined with the MLEM reconstruction method is a better choice among the coded-aperture families for the simulation of the 3D thick source like a breast phantom, as it has the capability to reconstruct a 3D breast phantom image from single scanning. Moreover, it reduces the noise of the MURA CA mask and can produce a better reconstructed image. The reconstructed images are evaluated by measuring the contrast-to-background ratio, peak signal-to-noise ratio, and mean square error.

## 5. Future Work

From the investigation of the mosaic MURA mask/antimask-based coded-aperture masks and by the inference of the results, it can be said that the mosaic MURA mask/antimask with the MLEM iterative reconstruction method has encouraging results in medical imaging, especially for small lesion detection, as it has reduced the significant effect of the sidelobes. By virtue of its improved performance, it can be used for other cancerous cells diagnosis like lung cancer, liver, etc. The proposed method may have the ability to detect the small lesion with a minimum dose because of the almost half-open area of the mask design. In future, if we have a proper computing facility, we can increase the size of the phantom and make it sensitive to the count coming out from the chest and other parts of the body.

## Figures and Tables

**Figure 1 diagnostics-12-02529-f001:**
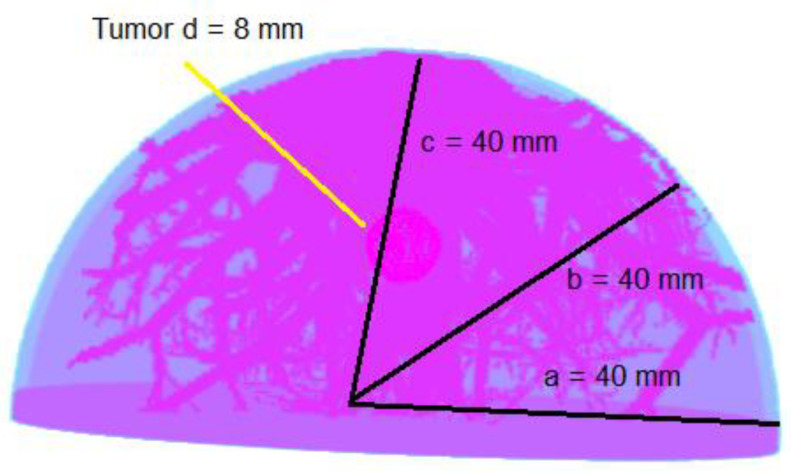
The anthropomorphic breast phantom with an 8 mm diameter tumor.

**Figure 2 diagnostics-12-02529-f002:**
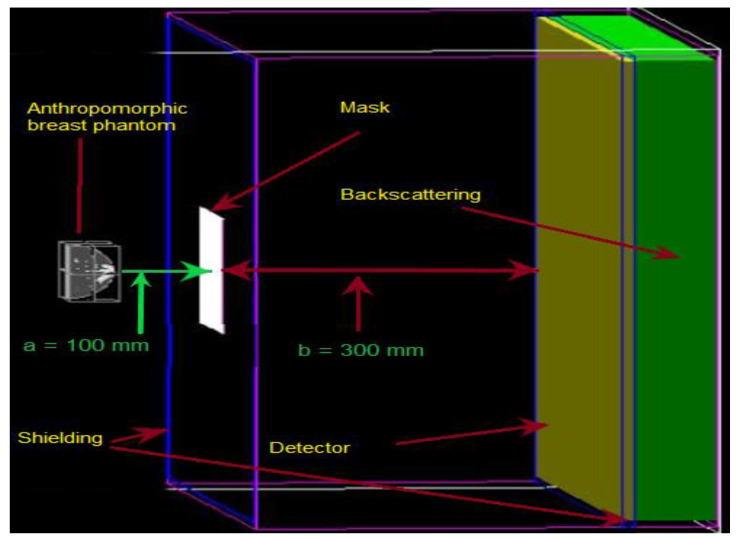
Visualization of simulated geometrical structure using GATE.

**Figure 3 diagnostics-12-02529-f003:**
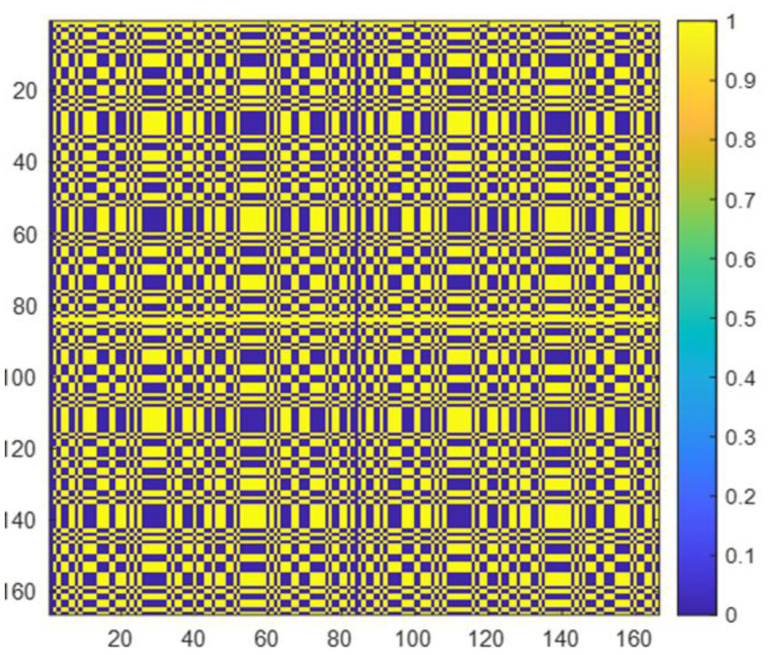
The mosaic MURA binary mask with 166 × 166 elements, blue color denotes o’s while yellow color denotes 1′s. The basic 83 × 83 MURA pattern was used to build this mosaic configuration.

**Figure 4 diagnostics-12-02529-f004:**
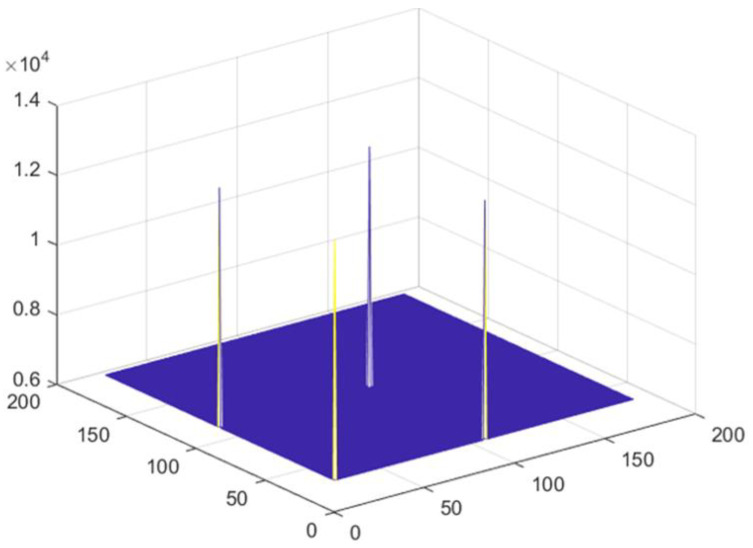
Response function of mosaic mask.

**Figure 5 diagnostics-12-02529-f005:**
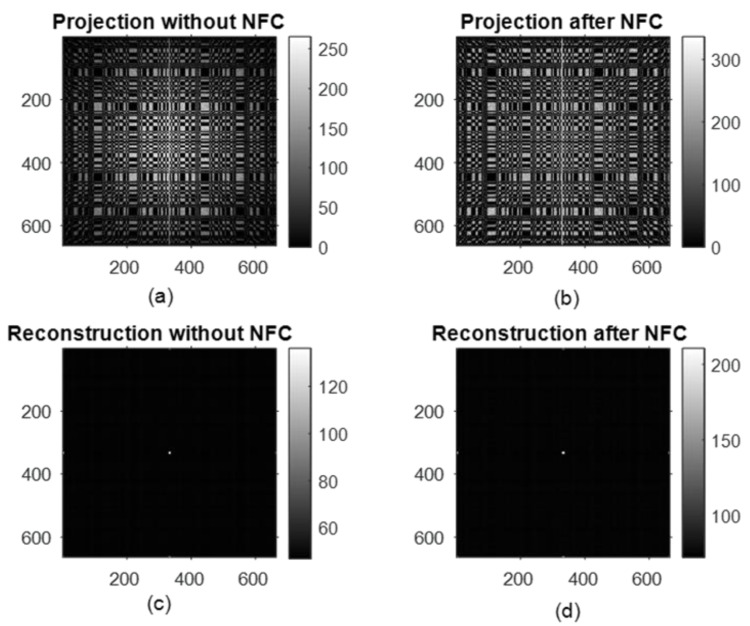
Projection of the simulated point source (**a**), corrected projection after near-field correction (**b**), reconstructed image of the point source without correction (**c**) and with correction (**d**).

**Figure 6 diagnostics-12-02529-f006:**
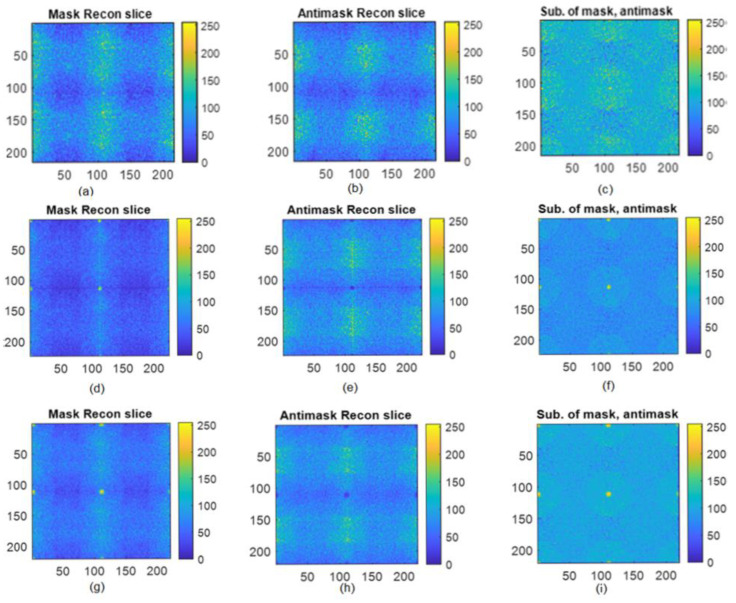
Reconstructed slice of an 8 mm tumor from different depths: (**a**) Breast phantom reconstructed slice at 12.5% depth of tumor using mask, (**b**) Breast phantom reconstructed slice at 12.5% depth of tumor using antimask, (**c**) difference of mask and antimask, (**d**) Breast phantom reconstructed slice at 25% depth of tumor using mask, (**e**) Breast phantom reconstructed slice at 25% depth of tumor using antimask, (**f**) difference of mask and antimask at 25% depth of tumor, (**g**) Breast phantom reconstructed slice at 50% depth of tumor using mask, (**h**) Breast phantom reconstructed slice at 50% depth of tumor using antimask, (**i**) difference of mask and antimask at 50% depth of tumor.

**Figure 7 diagnostics-12-02529-f007:**
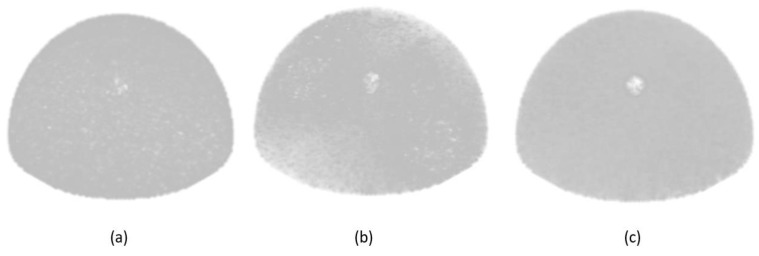
Reconstructed images of 3D breast with an 8 mm lesion: (**a**) using MURA mask/antimask, (**b**) using mosaic MURA mask, and (**c**) using mosaic MURA mask/antimask.

**Figure 8 diagnostics-12-02529-f008:**
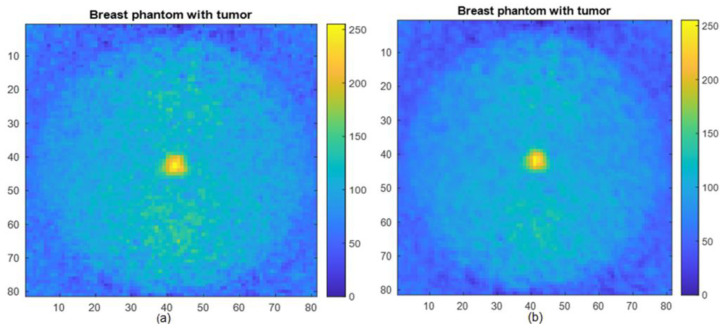
Reconstructed image with an 8 mm lesion: (**a**) central part, (**b**) central part added with partial copies along axial directions.

**Figure 9 diagnostics-12-02529-f009:**
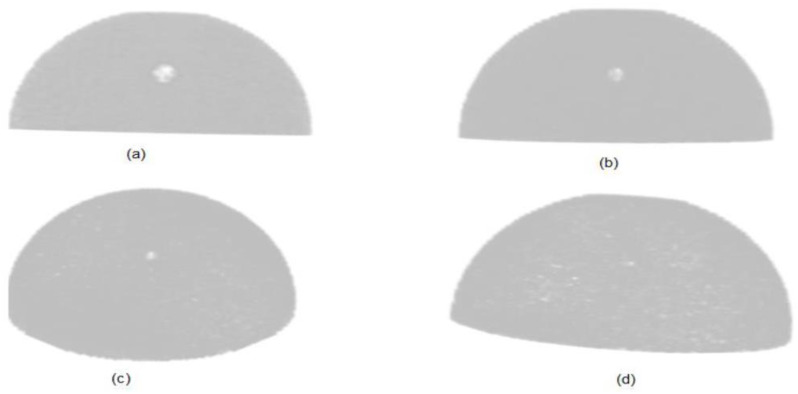
Breast phantom reconstructed images with (**a**) 8 mm tumor, (**b**) 6 mm tumor, (**c**) 4 mm tumor and (**d**) 3 mm tumor.

**Figure 10 diagnostics-12-02529-f010:**
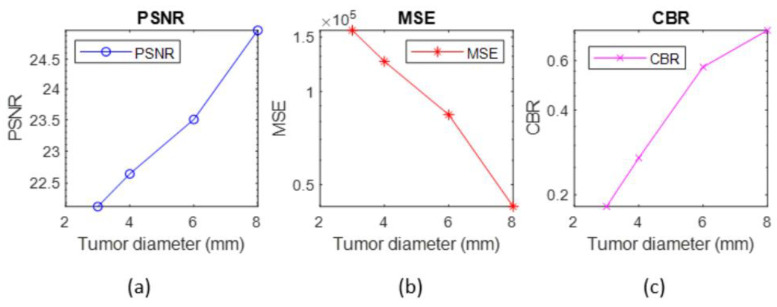
Breast phantom reconstructed images with (**a**) peak signal-to-noise ratio, (**b**) mean square error, (**c**) contrast-to-background ratio.

**Table 1 diagnostics-12-02529-t001:** Performance comparison among MURA mask/antimask, mosaic MURA mask and mosaic MURA mask/antimask.

	MURA Mask/ Antimask	Mosaic MURA Mask	Mosaic MURA Mask/Antimask
CBR	0.0348	0.2099	0.7692
PSNR	19.8098	21.2357	24.989
MSE	4.6154 × 10^5^	2.3935 × 10^5^	4.2498 × 10^4^

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
