# Peer review of "An Innovative Concept of a 3D-Coded Aperture Imaging System Proposed for Early Breast Cancer Detection"

_diagnostics, 2022, doi:10.3390/diagnostics12102529_

Round 1

Reviewer 1 Report

Detailed comments and a few English/typo edits can be found in the attached PDF. While the overall results appear to be correct, I found that the major assertion of a new distance correction technique is either not new, or inadequately described. Specifically, the near field correction embodied by equation 1 is a simple correction for the magnification of the mask pattern at the detector plane for a finite source distance. This is a well known correction and is essential to getting artifact free images in the near field. If there is more to the corrections applied by the authors, it is not described. Looking at Fig. 5, the difference between a and b does not seem to be magnification, but rather solid angle corrections toward the edge of the image. This is typically corrected to first order with, mask/anti-mask data collection, as first reported in the work of Accorsi and Lanza. In either case, more information is required.

I also find the use of the term mosaicked mask somewhat confusing. In my original reading of the paper, it seemed to be a new usage, although MURA masks are generally made of a 4-fold mosaic of the base pattern. I also note that the comments on line 247-251 on folding back data from outside the basic pattern is also not a new concept. For instance see Ziock et al, IEEE Trans. Nucl. Sci., Vol 60, p 2237 ; Braverman et al, IEEE Trans. Nucl. Sci.  Vol 62, 1405, and references therein.

Reference 9 is not really appropriate since it deals with optical light and encoding of a focussing image together with coded apertures.

The statements lines 64 to 65 that an MURA is best because of minimum side lobes and better spatial resolution are not correct. Other classes of coded apertures (URA, HURA, pseudo-random) all have zero side lobes, while the spatial resolution is the same for all of these as well.

Line 67, I do not understand the comment that the mosaicked mask reduces the inherent noise of the MURA. As stated above, this is the typical usage of MURA masks. If the reference is to folding back detector portions outside the primary field of view, this does not remove inherent noise to the MURA, but improves the counting statistics and hence the statistical noise in the image.

Fig. 2 seems to indicate that the single simulated view is toward the chest. If that is the case, does the simulation include background radiation from the rest of the patient (e.g. not just from the rest of the breast)? If not, then this must be included, otherwise please update the diagram and explain how radiation from the rest of the patient is shielded.

Lines 131-133, please provide the spatial and energy resolution values used in the simulation, and any other parameter settings. These could be given as a table.

Lines 207 & 208 What is the reconstruction method used to get these values.

Line 301 and 302, The statement that the MURA is a better choice is not supported. This can only be said if other coded apertures are tried and a comparison made.

Line 312, This statement is also not supported by the work done. There are no results with side-lobes that have been shown to go away wit the M-MLEM approach.

Line 316-317. It is not clear what background subtraction is planned. The mask/anti-mask technique is a form of background subtraction.

Finally, the paper would benefit from a comparison of these results to other coded-aperture results reported in the medical literature.

Author Response

Response to Reviewer 1 Comments

  1. Detailed comments and a few English/typo edits can be found in the attached PDF. While the overall results appear to be correct, I found that the major assertion of a new distance correction technique is either not new, or inadequately described. Specifically, the near field correction embodied by equation 1 is a simple correction for the magnification of the mask pattern at the detector plane for a finite source distance. This is a well known correction and is essential to getting artifact free images in the near field. If there is more to the corrections applied by the authors, it is not described. Looking at Fig. 5, the difference between a and b does not seem to be magnification, but rather solid angle corrections toward the edge of the image. This is typically corrected to first order with, mask/anti-mask data collection, as first reported in the work of Accorsi and Lanza. In either case, more information is required.

We developed a Matlab code using Accorsi and Lanza method to perform the near-field correction, and it has been more clearly mentioned in the revised manuscript.

  1. I also find the use of the term mosaicked mask somewhat confusing. In my original reading of the paper, it seemed to be a new usage, although MURA masks are generally made of a 4-fold mosaic of the base pattern. I also note that the comments on line 247-251 on folding back data from outside the basic pattern is also not a new concept. For instance see Ziock et al, IEEE Trans. Nucl. Sci., Vol 60, p 2237 ; Braverman et al, IEEE Trans. Nucl. Sci.  Vol 62, 1405, and references therein.

The term has the same meaning as mosaic, and now it has been replaced with the term mosaic.

  1. Reference 9 is not really appropriate since it deals with optical light and encoding of a focussing image together with coded apertures.

This reference 9 has been removed and new, more relevant references are added that discuss the more introductory detail of the coded aperture pinhole imaging.

  1. The statements lines 64 to 65 that an MURA is best because of minimum side lobes and better spatial resolution are not correct. Other classes of coded apertures (URA, HURA, pseudo-random) all have zero side lobes, while the spatial resolution is the same for all of these as well.

The statement has been updated in the revised manuscript.

  1. Line 67, I do not understand the comment that the mosaicked mask reduces the inherent noise of the MURA. As stated above, this is the typical usage of MURA masks. If the reference is to folding back detector portions outside the primary field of view, this does not remove inherent noise to the MURA, but improves the counting statistics and hence the statistical noise in the image.

This has been addressed using a 6mm lesion simulation with MURA mask antimask, mosaic mask, and mosaic mask antimask. Also, a comparison table is added in the result section.

  1. 2 seems to indicate that the single simulated view is toward the chest. If that is the case, does the simulation include background radiation from the rest of the patient (e.g. not just from the rest of the breast)? If not, then this must be included, otherwise please update the diagram and explain how radiation from the rest of the patient is shielded.

We are only providing the proof of concept that the 3D reconstruction of the breast phantom is possible with coded aperture imaging. Therefore, only the breast is modeled for the simulation, and adding the contribution of the chest and background will be the subject of future research. Right now, we are interested in reconstructing a 3D anthropomorphic breast phantom using a coded aperture mask. Therefore, the radiation contribution other than the breast is eliminated, and only radiations from the breast are considered. It has been added to the manuscript

  1. Lines 131-133, please provide the spatial and energy resolution values used in the simulation, and any other parameter settings. These could be given as a table.

The values of spatial resolution and energy resolution were mentioned in the revised manuscript.

  1. Lines 207 & 208 What is the reconstruction method used to get these values.

MLEM reconstruction method is used for the reconstruction of point source before and after the near-field correction.

  1. Line 301 and 302, The statement that the MURA is a better choice is not supported. This can only be said if other coded apertures are tried and a comparison made.

The comparison is made among MURA mask antimask, mosaic MURA mask, and the mosaic MURA mask antimask. It has been proved that the mosaic MURA mask antimask has enhanced the quality of the reconstructed images.

  1. Line 312, This statement is also not supported by the work done. There are no results with side-lobes that have been shown to go away wit the M-MLEM approach.

The simple convolution-based decoding has side-lobes due to the decoding noise and other near-field imaging artifacts. Moreover, with increasing number of iterations, the side-lobes further reduce until convergence. There is one below an example of decoding point source using convlotuion and MLEM algorithm. The image profile obtained from MLEM and convolution are given below. In MLEM, the image is smooth; although the peak is low, the resolution at full-width-half maximum is better for MLEM reconstructed image.

  1. Line 316-317. It is not clear what background subtraction is planned. The mask/anti-mask technique is a form of background subtraction.

Yes, the mask antimask is a type of background subtraction. But there is another way of doing background subtraction under progress, which will be unveiled soon.

  1. Finally, the paper would benefit from a comparison of these results to other coded-aperture results reported in the medical literature.

The coded aperture-based 3D breast phantom reconstruction is a novel technique, and there is not enough literature on 3D breast reconstruction to compare.

Reviewer 2 Report

 The authors demonstrates coded aperture imaging to  reconstruct 3D images of tumors within breast phantoms using a mask-anti mask scheme combined with Maximum likelihood maximization algorithm. The simulation and experimental results have been clearly presented in the paper and properly analyzed. I believe this manuscript can be published in the journal with some modifications and changes. My comments/suggestions are as follows:

1.    Figure 2 should be changed and more accurately made with proper labelling of the distances.

2.      Analysis for the mask and anti mask scheme is missing. It should be shown how it can reduce the noise.

3.      I would like to know about the filed of view and the angular resolution of the imaging technique.

4.      What was the reference image with which the reconstructed image was compared to calculate the figure of merit, and if any blind figure of merit can be analyzed ?

5.      How is the reconstruction different from reference27(Park et al) and if any other reconstruction technique was applied or not?

Author Response

 Response to Reviewer 2 Comments

  1. Figure 2 should be changed and more accurately made with proper labelling of the distances.

The figure is updated with a pointing arrow and color shades. Moreover, further elaboration of figure 2 regarding background shielding is also discussed.

  1. Analysis for the mask and anti mask scheme is missing. It should be shown how it can reduce the noise.

A table containing the simulation results of 8mm lesion using MURA mask antimask, mosaic MURA mask, and mosaic MURA mask antimask has been added. Moreover, figure 6, shows where some slices of 1mm thickness are reconstructed with mask and antimask.

  1. I would like to know about the filed of view and the angular resolution of the imaging technique.

The fully coded field of view is used for the breast phantom reconstruction, and it has been added in the first section of materials and methods.

  1. What was the reference image with which the reconstructed image was compared to calculate the figure of merit, and if any blind figure of merit can be analyzed ?

To calculate the PSNR, MSE the true source distribution of the breast phantom, is used. For result comparison, a table is added to further the results of mask antimask of MURA and mosaic MURA.

  1. How is the reconstruction different from reference27(Park et al) and if any other reconstruction technique was applied or not?

We used dual reconstruction with mask and antimask combined with MLEM reconstruction, while Park et al. used log-maximization of the Poisson-like probability function. Moreover, he simulated a point source, line source, and a small surface where the reconstruction is only in 2D, planar